# Reactive Oxidative Species in Carotid Body Chemoreception: Their Role in Oxygen Sensing and Cardiorespiratory Alterations Induced by Chronic Intermittent Hypoxia

**DOI:** 10.3390/antiox14060675

**Published:** 2025-06-01

**Authors:** Rodrigo Iturriaga, Hugo S. Diaz

**Affiliations:** Instituto de Ciencias Biomédicas, Facultad de Ciencias de la Salud, Universidad Autónoma de Chile, Santiago 8910060, Chile; hugo.diaz@uautonoma.cl

**Keywords:** carotid body, chronic intermittent hypoxia, inflammation, oxygen sensing, obstructive sleep apnea, oxidative stress, ROS

## Abstract

The carotid body (CB) senses arterial PO_2_, PCO_2_, and pH levels, eliciting reflex responses to maintain cardiorespiratory homeostasis. Chronic intermittent hypoxia (CIH), the hallmark of obstructive sleep apnea, elicits autonomic and cardiorespiratory alterations that are attributed to an enhanced CB chemosensory responsiveness to hypoxia, which in turn activates neurons and glial cells in the nucleus of the tractus solitarius (NTS). Although the CB contribution to the CIH-induced pathological alterations is well-known, the underlying mechanisms are not fully understood. A growing body of new evidence suggests a crucial role for ROS in acute CB oxygen sensing, as well as in the potentiation of chemosensory discharge and the activation of the central chemoreflex pathway in CIH. Indeed, it has been proposed that acute hypoxia disrupts mitochondrial electron transport, increasing ROS and NADH in the chemoreceptor cells, which inhibit voltage-gated K^+^ channels, producing cell depolarization, Ca^2+^ entry, and release of excitatory transmitters. In addition, new evidence supports that the enhanced CB afferent discharge contributes to persistent CIH-induced cardiorespiratory alterations, likely triggering neuroinflammation in the NTS. Thus, in this review, we will examine the experimental evidence that supports the involvement of ROS in the acute O_2_ sensing process, and their role in the enhanced CB chemosensory discharges, the glial-related inflammation in the NTS, and the cardiorespiratory alterations induced by CIH.

## 1. Introduction

### 1.1. The Carotid Body Chemoreceptors

The carotid body (CB) is the main peripheral chemoreceptor that senses changes in the arterial blood pressure of oxygen (PO_2_), the arterial pressure of carbon dioxide (PCO_2_), and pH, eliciting ventilatory, cardiovascular, and humoral reflex responses to maintain homeostasis [1,2,3]. The CB is a sensory organ situated in the bifurcation of the common carotid artery. The basic structure of the CB is the glomoid consisting of clusters of chemoreceptors (glomus or type I) cells around the capillaries and surrounded by glial (Type II) cells. The chemoreceptor cells make synaptic contact with the nerve terminals of primary afferent neurons whose cell bodies are in the petrosal ganglion and project through the carotid sinus nerve to the nucleus of the tractus solitarius (NTS). The NTS orchestrates the reflex responses projecting to the ventral respiratory group (vRG) and the rostral ventrolateral medulla (RVLM) in the brainstem and the paraventricular nuclei (PVN) in the hypothalamus [4,5,6] (see Figure 1).

The transduction of the natural stimuli occurs in the chemoreceptor cells and is linked to the Ca^2+^-dependent release of one or more excitatory transmitters such as adenosine triphosphate (ATP) and acetylcholine (ACh) ([1,3,7,8]. Although the CB detects changes in respiratory gases and pH levels in the arterial blood, it is considered a poly-modal receptor since it senses other sensorial modalities (i.e., temperature, flow, osmolarity, and glucose). In addition to the well-known role played by the CB in respiratory homeostasis, the CB also contributes to autonomic, cardiovascular, hormonal, and renal responses [1,2,3]. In addition to the excitatory transmitters between the chemoreceptor cells and the nerve endings of the petrosal neurons, molecules such as dopamine, angiotensin II (Ang II), aldosterone, endothelin I (ET-1), leptin, insulin, and nitric oxide (NO) modulate the chemoreception process, producing effects on chemoreceptor cells and blood vessels [3,7,9,10,11]. Moreover, new experimental evidence suggests that the CB chemoreceptor cells sense pro-inflammatory molecules, being part of the afferent arm of the anti-inflammatory reflex [3,12,13]. Indeed, interleukin-1β (IL-1β), interleukin-6 (IL-6), and tumor necrosis factor alpha (TNF-α) and their receptors are constitutively expressed in the CB tissue and modulate the chemosensory process [3,10,14,15,16].

### 1.2. The Oxygen-Sensing Mechanisms in the Carotid Body

The current model for O_2_ sensing in the CB states that the transduction of hypoxic stimuli occurs in the chemoreceptor cells and is associated with the Ca^2+^-dependent release of ACh and ATP, which in turn increases the rate of discharge of primary petrosal neurons that projects to the NTS [3,7,17]. Historically, two predominant hypotheses were proposed to explain the oxygen sensing mechanism. The “metabolic hypothesis” stated that the O_2_ sensor is linked to the mitochondrial oxidative metabolism within the chemoreceptor cells. Conversely, the “membrane hypothesis” suggested that the O_2_ sensor is associated with K^+^ channels in the chemoreceptor cell membrane. According to the metabolic hypothesis, hypoxia inhibits the mitochondrial electron transport and/or the oxidative phosphorylation in chemoreceptor cells, thereby reducing ATP content or altering the ATP/ADP+Pi ratio, which subsequently elevates Ca^2+^ and releases the excitatory transmitters [18,19,20]. In contrast, the membrane hypothesis states that hypoxia reversibly inhibits voltage-dependent K^+^ (KvO_2_) channels [21] and background K^+^ (TASK) channels [22], leading to cell depolarization, entry of Ca^2+^ through L-type voltage-dependent Ca^2+^ channels, which prompts the release of transmitters.

A critical evaluation of these hypotheses revealed that none of them could explain the entire O_2_ sensing process. However, both mechanisms are potentially involved in O_2_ sensing, and new findings suggest that they may interact in the transduction process. In biological systems, molecules related to the redox oxidative mitochondrial metabolism play crucial roles in physiological function. Accordingly, it has been proposed that hypoxia impairs mitochondrial electron transport and/or oxidative phosphorylation increasing reactive oxygen species (ROS) levels [23,24,25] and/or decreasing ATP [26,27], which in turn control the opening of KvO_2_ and TASK K^+^ channels, respectively, eliciting membrane depolarization, entry of extracellular Ca^2+^, and the release of the excitatory transmitters [3].

### 1.3. Contribution of the Carotid Body to the Pathological Consequences of Autonomic-Related Diseases

The proposal that CB contributes to the pathological consequences of resistant hypertension, congestive cardiac failure, obstructive sleep apnea, and metabolic disorders received much attention in recent years [1,3,28,29,30,31,32]. The evidence supporting a significant role for CB in these diseases is compelling. An abnormally enhanced CB chemosensory responsiveness to hypoxia has been related to sympathetic-adrenal hyperactivity, a common characteristic of those diseases [1,29,30,32,33,34,35,36]. The mechanisms underlying the CB chemosensory potentiation are not completely understood. However, it has been proposed that oxidative stress and inflammation are involved in CB chemosensory potentiation (see abstract figure, [30,32,33,34,37,38,39]). Moreover, the ablation of the CB chemosensory input to the NTS reduced the sympathetic overactivity and the cardiorespiratory alterations in preclinical models of these autonomic-related pathologies, highlighting the role of the CB [3,32,39].

A growing body of new evidence shows that autonomic and cardiorespiratory alterations elicited by chronic intermittent hypoxia (CIH), the main feature of obstructive sleep apnea (OSA), are critically dependent on an abnormal heightened CB chemosensory input to the NTS, activating second-order neurons that project to the autonomic, cardiovascular, and respiratory nuclei in the brainstem and hypothalamus [39,40,41]. Furthermore, CIH produced oxidative stress and inflammation in the CB and the NTS, which is crucial for generating and maintaining the pathological consequences produced by CIH. Accordingly, the main goal of this review is to provide a balanced discussion on the contribution of ROS and pro-inflammatory molecules in CB function in health and diseases, and their contribution to the enhanced CB chemosensory responsiveness to hypoxia induced by CIH and the generation of cardiorespiratory and autonomic alterations in preclinical OSA models.

## 2. The Carotid Body Oxygen Sensing Process and ROS

### 2.1. The Mitochondria to Membrane Signaling Model for O_2_ Sensing in the Carotid Body

López-Barneo and colleagues advanced an innovative hypothesis to elucidate the O_2_ sensing mechanism in the CB, which integrates mitochondrial ROS and KvO_2_ channels in the microdomains between the mitochondria and the plasmatic membrane of chemoreceptors cells [21]. They proposed the “mitochondria to membrane signaling model for acute O_2_ sensing” that integrates mitochondrial metabolism and KvO_2_ channel function, based on the mitochondrial production of ROS and pyridine nucleotides (NADH) in response to acute hypoxia. According to their proposal, the O_2_ sensor is the mitochondrial complex IV, while the O_2_ sensing process depends on the ROS production in the complex I of the CB chemoreceptor cells [23,24,25,42]. The hypoxic inhibition of the mitochondrial electron transport chain increases the production of ROS and NADH in complex I of the mitochondria. They proposed that hypoxia increases the reduced state of mitochondrial complex IV by inducing the accumulation of electrons in the electron transport chain, which in turn increases the levels of reduced ubiquinone (CoQH2), slowing the activity of the mitochondrial complex I and raising ROS and NADH levels. This mechanism for ROS and NADH production is known as reverse electron transport and is produced when electrons from ubiquinol are transferred back to complex I, reducing NAD+ to NADH. This process generates a significant amount of ROS, including superoxide anion and hydrogen peroxide [43]. The increased levels of ROS and NADH acting at the microdomains between the mitochondria and the cell membrane of the chemoreceptor cells inhibit the conductance of the KvO_2_ channels, depolarizing the cell and eliciting the release of the excitatory transmitters [12,24,44]. Interestingly, oxidizing and/or reducing agents and ROS scavengers did not have appreciable effects on TASK K^+^ channels [45].

### 2.2. Unique Higher Oxygen Sensitivity of the Carotid Body Chemoreceptor Cells

Gao et al. (2017) [24] studied the gene expression profiles from CB chemoreceptor cells, adrenal chromaffin cells, and superior cervical ganglion neurons because these tissues display different O_2_ sensitivity and have a common lineage and are catecholaminergic cells. Interestingly, they found the presence of three atypical mitochondrial subunits (Ndufa4l2, Cox4i2, and Cox8b) and the upregulation of the hypoxic inducible factor (HIF2α) in the CB chemoreceptor cells but not in adrenal chromaffin cells or autonomic neurons [44]. COX4i2 is a nuclear-encoded isoform of the cytochrome c oxidase (complex IV), which catalyzes the electron transfer from reduced cytochrome c to molecular O_2_, contributing to the generation of the electrochemical gradient along the inner mitochondrial membrane. The presence of the mitochondrial subunit COX4i2 makes the cytochrome c oxidase more sensitive to hypoxia eliciting a backup of electrons in the electron transport chain and increasing the CoQH2/CoQ ratio, which in turn increases ROS and NADH production in mitochondrial complex I during hypoxia. The mitochondria from other cells have a low threshold for PO_2_ levels. Indeed, the cytochrome a3 displays a Km of < 1 mmHg in isolated mitochondria and one in the range of 1–5 mmHg in dissociated cells [46]. However, the CB chemoreceptor cells possess the ability to be activated at a much higher level of PO2 [47,48,49]. Lahiri et al. (1993) [48] measured microvascular PO_2_ in cat CB in vivo using an optical method based on the O_2_-dependent quenching of the phosphorescence emitted by a chromoporphine molecule. They found that the CB microvascular PO_2_ was in the order of 50 mmHg when the femoral arterial PO_2_ was 110 mmHg. In response to hypoxia, the increase in the CB chemosensory discharge begins at a PO_2_ of 40 mmHg in the CB vessels, while the arterial systemic PO_2_ was 90 mmHg. Since the chromoporphine molecules remained in the capillaries because they are associated with albumin, the actual PO_2_ in the CB tissue should be much lower [48].

Moreno-Domínguez et al. (2020) [17] found that the increases in NADH and ROS levels in the intermembrane mitochondrial space during hypoxia were strongly reduced in the knockout Cox4i2 mice, suggesting that ROS increases occur from an accumulation of electrons proximal from the mitochondrial complex IV. Fernandez-Aguera et al. (2015) [42] studied the hypoxic and hypercapnic ventilatory responses in a transgenic mouse without the Ndufs2 gene that encodes for the 49-kD subunit of the ubiquinone binding site for rotenone in the mitochondrial complex I in the CB chemoreceptor cells. They found that the hypoxia ventilatory response was abolished but not the response to hypercapnia in the knockout mice. Besides that, hypoxia failed to increase the dopaminergic secretion in chemoreceptor cells lacking Ndufs2l, but chemoreceptor cells showed normal responses to hypercapnia, hypoglycemia, and high extracellular K^+^ [42].

Gao et al. (2021) measured acute changes in ROS production in CB slices using redox-sensitive green fluorescent-protein probes and microfluorimetry [44]. They measured ROS during acute hypoxia in different subcellular compartments of mice CB chemoreceptor cells and assessed the effect of mitochondrial complex I on acute O_2_ sensing in the knockout mice deficient for subunit Ndufs2 in the mitochondrial complex I. They found that hypoxia increases ROS production in the mitochondrial intermembrane space but not in the mitochondrial matrix of wild-type mice, but ROS were completely absent in chemoreceptor cells from Ndufs2 knockout animals. Thus, these results suggest that ROS generated from the mitochondrial intermembrane space with functional complex I, but not from the mitochondrial matrix, acts as signaling molecules during acute hypoxia linking mitochondrial metabolism with the activity of KvO_2_ channels in the membrane of chemoreceptor cells. In summary, the available evidence supports the idea that ROS signals participate in the acute O_2_ sensing process in the CB chemoreceptor cells.

## 3. Chronic Intermittent Hypoxia Enhanced Carotid Body Chemosensory Discharges in Preclinical Models of Obstructive Sleep Apnea

### 3.1. Obstructive Sleep Apnea

Obstructive sleep apnea (OSA) is a growing public health problem worldwide. Indeed, 10% of adult men and 5% of women show an apnea/hypopnea index of 10 or more events per hour [50,51]. However, more recent estimates suggest that 930 million individuals worldwide suffer from OSA, with a higher prevalence in adult males [52]. OSA is associated with sleep fragmentation, somnolence, and cognitive dysfunction [53]. However, OSA is recognized as an independent risk factor for systemic hypertension and is associated with coronary disease, arrhythmias, stroke, and pulmonary hypertension [51,54,55]. Certainly, 50% of OSA patients develop hypertension, with a positive correlation between the number of episodic apnea/hypopnea and the prevalence of resistant hypertension [56,57]. OSA is characterized by cyclic events of partial or total airflow arrest during sleep due to the collapse of the upper airways. During an apnea or hypopnea episode, the resulting hypercapnia and hypoxia stimulate the CB, which increases the ventilatory motor output leading to negative thorax pressure, micro-arousals, and, finally, the restitution of the breathing airflow [51,54]. Among these disturbances produced by airflow occlusion, CIH is considered the main factor for the development of hypertension in OSA patients [51,54,58]. Furthermore, CIH is sufficient to elicit autonomic and cardiorespiratory alterations, including hypertension in rodents, the gold-standard preclinical model of OSA mimicking the pathological alterations found in OSA patients [34,59,60,61].

### 3.2. Chronic Intermittent Hypoxia Produces Carotid Body Chemosensory Potentiation

The cardiovascular consequences of OSA have been attributed to oxidative stress, inflammation, endothelial dysfunction, and sympathetic overflow [51,54,55,58,62]. Indeed, OSA patients and rodents exposed to CIH develop sympathetic hyperactivation, increased breathing instability and breathing–sympathetic coupling, reduction in cardiac baroreflex sensitivity, and heart rate variability alterations [1,3,63,64,65,66]. The enhanced ventilatory and pressor responses to hypoxia found in OSA patients were attributed to a potentiated hypoxic peripheral chemoreflex drive, suggesting that the CB contributes to the generation of those alterations [65]. Moreover, Fletcher et al., 1992 [59] found further evidence for the crucial role of CB in CIH-induced hypertension. They found that CB denervation prior to the CIH exposure for 35 days prevented the increases in arterial blood pressure in rats [59]. Although this seminal finding supports a primary role for the CB in CIH-induced hypertension, the role of the CB in the cardiorespiratory changes induced by CIH has been seriously considered only in the past decade. New experimental findings have shown that the CB is a crucial component in the generation of autonomic and cardiorespiratory alterations caused by CIH. Indeed, neural recordings of CB chemosensory discharges showed that CIH selectively increases baseline CB chemosensory discharge in normoxia and enhances chemosensory responses to hypoxia in animal models of OSA [34,37,63,67,68]. Del Rio et al., 2016 [69] studied whether autonomic and cardiorespiratory alterations depend on the CB integrity, by eliminating the CB input to the NTS in free-moving rats exposed to mid-term CIH for 21 days. They found that CIH increased the arterial blood pressure (15 mmHg) measured by indwelling telemetry, enhanced the ventilatory hypoxic response, produced autonomic imbalance with sympathetic prevalence, decreased the baroreflex gain, and increased the number of arrhythmias. Following 21 days of CIH, rats underwent bilateral CB ablation that promptly normalized the hypertension and the enhanced ventilatory response to hypoxia and restored the normal heart rate variability and the baroreflex sensitivity, even when rats were kept in CIH for one more week [69]. More recently, Pereyra et al., 2025 [70] studied if the persistence of cardiorespiratory alterations found in long-term CIH depends on the enhanced CB inputs to the NTS. They denervated the CBs from mice exposed for 50 days to CIH and kept the animals in CIH for another 10 days. Their results showed that the increased arterial blood pressure and the enhanced hypoxic ventilatory response were abolished by CB denervation. These results highlighted the role played by the CB in the progression of cardiorespiratory alterations and supported the idea that the surgical ablation of the CB may be useful in restoring normal cardiorespiratory function in humans. The CB ablation has been used in pilot human studies as a therapeutic treatment for resistant hypertension and heart failure-induced sympathetic hyperactivity, but caution is required before the use of bilateral CB ablation in obstructive sleep apnea that abolished ventilatory responses to hypoxia [see 3 and 14 for review]. In summary, the current evidence supports that an enhanced CB chemosensory input to the NTS is the main factor for generating and maintaining sympathetic overflow and cardiorespiratory alterations in free-moving rodents exposed to CIH, the gold-standard preclinical model of OSA.

## 4. Carotid Body Chemosensory Potentiation Induced by CIH: The Role of Oxidative Stress

### Oxidative Stress Induced by Chronic Intermittent Hypoxia the Carotid Body and the Chemoreflex Neural Pathway

ROS and reactive nitrogen species (RNS) have been proposed as mediators for autonomic and cardiorespiratory alterations in OSA patients [58,71,72] and animals exposed to CIH [34,37,64,73,74,75]. Oxidative stress is produced by the disparity between the increased production of ROS and the action of antioxidant defenses, leading to an increment of ROS that may produce oxidative damage in cells and tissues. ROS are produced by the mitochondrial metabolism, the NADPH oxidase, the xanthine oxidase, the nitric oxide synthase, and peroxisomal constituents. ROS are unstable and highly reactive molecules that contribute to the generation of pathological effects of CIH in preclinical OSA models. This idea is strongly supported by the fact that different antioxidants, such as ascorbic acid, n-acetylcysteine, tempol, and superoxide dismutase mimetics administered prior to or concomitantly with the CIH stimuli, prevented the cardiovascular, autonomic, and ventilatory alterations [68,73,76,77,78,79,80] (graphical abstract).

Peng & Prabakar (2003) [37] proposed that the superoxide radical contributed to the potentiation of the rat CB chemosensory responses to hypoxia induced by CIH. They found that pretreatment of rats for 10 days before and during the exposure to CIH for 10 days with the superoxide dismutase mimetics manganese (III) tetrakis (1-methyl-4-pyridyl) porphyrin pentachloride (MnTMPyP) prevented the CB chemosensory potentiation and oxidative stress. In addition, they found that CIH decreased the activity of the aconitase enzyme in the CB and in the activity of the complex I, suggesting that the mitochondria function is affected by CIH, being a potential source of ROS production during CIH. In a later study, Peng et al. (2009) [68] found evidence that ROS generated by NADPH oxidase (NOX) contributes to the CIH-induced CB potentiation since acute hypoxia stimulus increases NOX activity in chemoreceptor cells CB from rats exposed for 10 days to CIH. They also found that the enhanced CB chemosensory discharges and the responses to hypoxia were reduced by NOX inhibition and were not present in knockout NOX mice. Figure 2 compares the origin and effects of ROS on acute O_2_ sensing and during CIH. In the acute hypoxic transduction process, ROS are derived from the complex I of the electron transport chain [17,42,44]. In CIH, ROS are not only produced in complex I, but also by the NADPH oxidase [37,68].

The progression of the oxidative stress in the CB has been addressed by measuring the accumulation of the immunoreactive 3-nitrotyrosine (3-NT) in rats exposed to CIH [76,81]. Superoxide reacts with NO to produce radical peroxynitrite that nitrates tyrosyl groups in proteins altering its function through the generation of 3-NT residues. CIH produces a progressive increase in 3-NT in the CB tissue from 3 to 7 days of CIH, which is maintained during 28 days of CIH exposure. The time course of 3-NT accumulation paralleled the potentiation of the CB baseline discharge in normoxia and the enhanced chemosensory response to hypoxia, the potentiation of the ventilatory response to hypoxia and hypertension [76]. Concomitant treatment with ascorbic acid in the drinking water during CIH exposure reduced the CB oxidative stress, normalized the enhanced CB chemosensory discharges, and prevented the potentiated ventilatory responses to hypoxia and elevated arterial blood pressure [76]. Moya et al. (2016) [73] found that the administration of Ebselen, a specific peroxynitrite scavenger, during CIH for 7 days reduced the 3-NT levels, normalizing the chemosensory response to hypoxia and hypertension. Morgan et al. (2016) [82] measured the responses to acute hypoxia in free-moving rats exposed to 14 to 21 days of CIH, with or without the concomitant administration of the xanthine oxidase inhibitor allopurinol, allopurinol plus the Ang II receptor antagonist Losartan, and the NADPH oxidase inhibitor apocynin. They found that all pharmacologic interventions effectively eliminated the enhanced ventilatory response to acute hypoxia, while 3-NT accumulation in the CB was attenuated by allopurinol plus losartan and by apocynin, but was unaffected by allopurinol. Accordingly, the ventilatory chemoreflex activation was attenuated by the inhibition of NADPH oxidase, xanthine oxidase, and the inhibition of xanthine oxidase combined with the blockade of angiotensin II receptors, via antioxidant mechanisms. Nevertheless, it is not possible to attribute all the effects to inhibition at the CB level because these antioxidants were given in a systematic manner. Thus, it is not possible to preclude any effect on the central components of the CB chemoreflex pathways such as the NTS and the PVN.

It is well known that the accumulation of ROS increased the inducible hypoxic factor HIF-1α and reduced the HIF-2α levels in the CB, promoting a pro-oxidant environment in the CB [38,60,83]. Accordingly, the elevated HIF-1α levels induce the expression of the pro-oxidant enzyme NOX and the reduced HIF-2α level lowers the expression of the antioxidant enzyme manganese-dependent superoxide dismutase (MnSOD) [38,84]. Therefore, changes in the HIF-1α/HIF-2α ratio may contribute to maintaining the oxidative stress in the CB due to a reduced MnSOD activity and an increased NOX [84]. Kumar et al. (2015) [85] reported that CIH for 7 days increased in the adrenal medulla aconitase activity and produced an imbalance of the ratio HIF-1α/HIF-2α, which in turn increased the expression of NOX and reduced MnSOD, leading to an increased ROS production. Interestingly, these alterations in the adrenal medulla induced by CIH were abolished by the CB ablation, suggesting that CIH-induced oxidative stress in the adrenal gland depended on the activation of the CB chemoreflex pathway. Moya et al. (2018) [86] studied whether CIH-induced peroxynitrite may nitrate MnSOD and the cytoplasmic superoxide dismutase (CuZnSOD) decreasing their enzymatic activity in the rat CB, superior cervical ganglion, and adrenal gland. They found that exposure to CIH for 7 days increased 3-NT levels in the CB and adrenal gland and raised MnSOD protein levels in the CB and in the adrenal cortex, but not in the whole adrenal medulla and the superior cervical ganglion. CIH nitrated MnSOD in the CB and adrenal medulla, decreasing its activity while CuZnSOD levels remained unchanged in all tissues studied. The ascorbic acid treatment prevented all changes. Accordingly, it is likely that the increase in MnSOD activity in the CB resulted from the increase in MnSOD protein levels following 7 days of CIH. Interestingly, MnSOD protein levels increased everywhere except in the adrenal medulla. Nitration inhibited MnSOD, explaining why CIH reduced MnSOD activity in the rat adrenal medulla. Thus, it is likely that CIH-produced nitro oxidative stress in the CB may depend on the mitochondrial accumulation in complex 1 and NOX activation, which was not entirely compensated by MnSOD and CuZnSOD.

Not all the changes in the CB are attributed to CIH-induced oxidative stress. Indeed, CIH potentiated the CB chemosensory responsiveness to hypoxia without changing the CB size. Indeed, CIH did not change the volume of the rat CB or the number and size of chemoreceptor cells, but increased the VEGF levels and enlarged blood vessels, while the vessel count remained unchanged [87]. Ascorbic acid prevents the CB chemosensory potentiation but does not affect vascular enlargement or elevated VEGF levels, indicating that hypoxia directly causes vascular enlargement in the CB [87].

## 5. Carotid Body Chemosensory Potentiation Induced by CIH: Role of Pro-Inflammatory Molecules

### Oxidative Stress and Inflammation in the Carotid Body and the Chemoreflex Neural Pathway Induced by Chronic Intermittent Hypoxia

It is not completely known how ROS may enhance CB chemosensory baseline in normoxia and the chemosensory responses to hypoxia during CIH. It is likely that the augmented ROS production during CIH may inhibit the opening of KvO_2_ channels in the membrane of chemoreceptor cells (Figure 2). On the other hand, oxidative stress activates the nuclear factor κB (NF-κB), the activator protein 1 (AP-1), and HIF-1α in the CB [60,83], which are crucial transcription factors that orchestrate the expression of many genes involved in inflammation like ET-1 [88,89], Ang II [90], and inducible nitric oxide synthase iNOS [91], which contribute to enhanced CB chemosensory responses to intermittent hypoxia. CIH also increases the levels of IL-1β, IL-6, and TNF-α in the rat CB [81,91,92]. Del Rio et al., 2011 [91] found a progressive increase in TNF-α, IL-1β, and iNOS levels in the rat CB 14 days of CIH, while ET-1 showed a transient increase at 1 week of CIH that promptly returned to basal levels. Interestingly, the administration of ascorbic acid abolished the increased TNF-α and IL-1β levels in the CB, indicating that CB inflammation depends primarily on oxidative stress [81]. Del Rio et al. [93] found that ibuprofen prevented the overexpression of pro-inflammatory cytokines in the CB, increased arterial blood pressure, and the ventilatory response to hypoxia. Furthermore, ibuprofen reduced the increased CB chemosensory baseline in normoxia but failed to block completely the enhanced CB chemosensory responses to hypoxia. Interestingly, systemic treatment of ibuprofen prevented the neuronal activation (c-fos positive cells) in the NTS on rats exposed to CIH for 21 days, suggesting that the neuronal activation depends on the activation of inflammatory pathways in the NTS [93]. It is worth mentioning that CIH increases the levels of IL-1β, IL-6, and TNF-α in the CB [91] and in the NTS [40], suggesting that pro-inflammatory cytokines in the NTS may contribute to the maintenance of cardiorespiratory alteration in CIH-exposed animals. Similarly, Popa et al. (2011) found that the administration of ibuprofen in rats exposed to sustained hypoxia for 10 days reduced the increase in IL-1β and IL-6 in the NTS and reduced the enhanced ventilatory response to hypoxia, indicating that these cytokines are crucial for the ventilatory response to sustained hypoxia [94].

## 6. Chronic Intermittent Hypoxia-Induced Activation of the Carotid Body Chemosensory Pathway

### 6.1. Chronic Intermittent Hypoxia Activates Neurons and Glial Cells in the Nucleus of the Tractus Solitarius

The NTS is the primary site for the integration of respiratory and cardiovascular inputs, including the CB chemoreflex [4,5,6,95]. The petrosal neurons that innervate the CB chemo-receptor cells project to the caudal section of the NTS, specifically to the dorsal, medial, and commissural sub-nuclei. Second- and third-order neurons in the NTS project to the RVLM, where pre-sympathetic neurons innervate pre-ganglionar sympathetic neurons. It is known that CIH produces oxidative stress, inflammation, and impaired neuronal and glial function in the brainstem, producing the dysregulation of gene expression related to synaptic transmission, ion channels, and inflammatory responses [40,96,97,98,99,100]. A meta-analysis robustly showed that CIH, independently of other confounders, has a strong effect on the brain by inducing oxidative stress, inflammation, and apoptosis in rodent models. Indeed, CIH induced oxidative stress, increased NF-κB, NADPH oxidase, TNF-α, iNOS, and decreased SOD levels [101]. CIH induced oxidative stress and neuronal activation in the NTS, the RVLM, and the PVN [93,102,103,104]. Since the CIH-induced increases in FosB in the RVLM are reduced by systemic treatment with the superoxide mimetic agent MnTMPyP [64], it is plausible that neuronal activation may result from CIH-induced oxidative stress. However, it is also possible that the CB-mediated activation of the central chemoreflex pathway is responsible for the neuronal activation in the NTS and the RVLM. Furthermore, CB neurotomy performed before the onset of the CIH exposure for 30 days in rats prevented the oxidative stress in the NTS and RVLM and hypertension [38]. Accordingly, the available evidence supports that the activation of the CB chemosensory pathway is responsible for the oxidative stress in the NTS and RVLM.

The glutamatergic synapses between the petrosal chemosensory neurons and the second-order neurons in the NTS are potentiated by CIH. Indeed, CIH enhances the glutamatergic transmission in the NTS, activating NMDA and AMPA currents [105]. CIH increased excitatory synaptic activity eliciting long-term potentiation, an effect attributed to the activation of TRPV1 channels [106]. The activation of the neurons in the NTS required the participation of glial cells. It is known that astrocytes in the NTS participate in the tripartite synapse and contribute to neuronal activation and plasticity by the uptake of molecules from the synaptic cleft and by releasing gliotransmitters (i.e., glutamate, ATP) [4,107,108]. Glial cells modulate the synaptic transmission in the NTS during acute and sustained hypoxia by changing the expression and/or function of the tripartite synapse components [107,109,110]. Microglia are the dominant pro-inflammatory cells in the central nervous system. In response to oxidative stress, microglia display morphological changes and secreted pro-inflammatory cytokines playing a significant role in the pathological cardiorespiratory consequences in several models of neuroinflammation in the NTS [3,111,112]. CIH activates microglia producing oxidative stress and inflammation and the release of excitatory neurotransmitters [113,114]. Tadmouri et al. (2014) [114] studied the activation of glial markers in the NTS during sustained hypoxia in mice and found that astrocytes were activated first by hypoxia, followed by microglia. Minocycline that suppressed microglial activation also decreased astrocyte activation and reduced the basal increase in ventilation.

### 6.2. Role of Nucleus of the Tractus Solitarius Resident Glial Cells on the Inflammation Induced by Chronic Intermittent Hypoxia

Recently, Pereyra et al. [70] studied the morphological activation of microglial cells and astrocytes in the NTS of mice exposed to long-term CIH for 60 days [70]. The Sholl analysis showed a significant transient increase in astrocyte morphological complexity at 7 days of CIH, featured by a transient increase in the number of process intersections. The astrocyte activation was accompanied by increases in branch length without changes in the total number of branches. The transient changes in astrocyte morphology were followed by a progressive normalization at 14 days of CIH that persisted until 60 days of CIH. Contrarily, a significant increase in the number of microglia process intersections occurred later during the CIH progression, becoming significant after 14 days of CIH and remaining elevated up to 60 days of CIH. The increase in the number of intersections corresponded with increases in branch length and the total number of branches. Additionally, microglial cell complexity increased after 14 days, indicated by an elevated cell volume that persisted until 60 days of CIH. CIH triggers distinct activation patterns in astrocytes and microglial cells in the NTS. Therefore, early astrocyte activation in the NTS suggests involvement of astrocytes in the initial progression of the cardiorespiratory dysfunction induced by the potentiation CB chemoreceptor discharges, which elevated blood pressure, and the ventilatory response to hypoxia after 7 days of CIH. The late microglial activation contributes to the generation of a pro-inflammatory niche in the NTS. This is supported by the fact that long-term CIH for 60 days increased RNA levels of TNF-α, IL-6, and IL-1 in the mice NTS [70].

Yamamoto et al. (2012) [115] reported that short-term CIH for 7 days did not modify the protein levels of the astrocyte activity marker glial fibrillary acidic protein in the rat NTS. This result was supported by the observation that bilateral microinjections of Fuorocitrate into the NTS did not alter baseline respiratory and sympathetic discharge in juvenile rats exposed to CIH for 10 days ([116]. However, Fluorocitrate is a metabolic toxin that is selectively taken up by astrocytes and produces membrane depolarization by disruption of the Na+/K+ ATPase [117]. Astrocytes participate in the transmitter and ionic control during enhanced neuronal discharge activity. Indeed, glutamate and K^+^ accumulated in the tripartite synaptic cleft are taken up by astrocytes through ex-citatory amino-acid transporters (EAAT), K^+^ channels, or Na^+^/K^+^ pumps, respectively. Martinez et al. (2020) [118] found that astrocytes are engaged in the activation of neurons in the NTS from rats exposed to CIH for 7–14 days. They found that CIH reduced the expression and levels of EAAT in the rat NTS astrocytes, increased the extracellular glutamate levels, and produced a large spontaneous synaptic activity and neuronal depolarization. Taken together, these results support a regulatory role for glial cells in the NTS synapse during both sustained and CIH. Thus, it is likely that astrocytes may contribute to the generation of cardiorespiratory alterations in the preliminary stages of CIH.

### 6.3. Crucial Role of the Carotid Body in the Glial Cell Activation and Inflammation Elicited in the Nucleus of the Tractus Solitarius by Chronic Intermittent Hypoxia

Pereyra et al., 2023 [119] studied if acute astrocyte inhibition in the NTS using Designer Receptor Exclusively Activated by Designer Drugs (DREADD) reduced the autonomic alterations and elevated blood pressure in rats exposed to CIH for 21days. They exposed male rats to normoxic conditions for 7 days and then to CIH for 21 days. At 7 days of CIH, in anesthetized rats, both NTS were stereotaxically injected with an adeno-associated virus containing DREADD-inhibitory (Gi) under the control of the GFAP promoter for selective astrocyte inhibition. At 21 days of CIH, hemodynamic parameters were recorded before and after the acute inhibition of NTS astrocytes with clozapine N-oxide. Pereyra et al., 2023 [119] found that acute chemogenic inhibition of NTS astrocytes following 21 days of CIH reduced the hypertension, enhanced pressor response to hypoxia, and restored the heart rate variability.

Since CIH enhances the CB chemosensory response to hypoxia, which in turn potentiates the hypoxic central chemoreflex pathway [39,40,63,93], Pereyra et al., 2025 [70] studied whether long-term CIH-induced cardiorespiratory alterations depend on the CB inputs to the NTS and in what extent glial cells activation and neuroinflammation in the NTS contribute to these changes. Both CBs were denervated in mice at 50 days of CIH exposure and animals were maintained under CIH for 10 more days. They found that CB denervation was sufficient to normalize the elevated arterial blood pressure and the enhanced hypoxic ventilatory response, but did not completely restore the CIH-induced irregular breathing pattern, suggesting that plastic changes may occur in central respiratory nuclei, particularly in the respiratory rhythm generator preBötzinger nucleus [120]. Remarkably, CB denervation did not alter the astrocyte morphology but produced a marked reduction in the increased number of microglia processes intersection and a reduced number in the total number of microglia branches. Importantly, CB denervation in CIH mice resulted in significant reductions in TNF-α and IL-6, but not completely reduced IL-1β [70,118].

Thus, the experimental data support that CIH, the feature of obstructive sleep apnea, causes autonomic and cardiorespiratory alterations (i.e., sympathetic overflow, hypertension, and potentiated ventilatory responses to hypoxia) linked to oxidative stress and neuroinflammation in the CB and the NTS. Furthermore, enhanced CB chemosensory discharge may contribute to the generation and persistence of the CIH-induced alteration triggering a pro-inflammatory niche in the NTS. Figure 3 shows the proposed pathophysiological mechanisms for the potentiation of CB chemosensory, and glial and neuronal activation in the NTS during CIH. It is likely that microglial activation in the NTS may contribute to the increased cytokine levels in the NTS. Early astrocyte activation in the NTS during CIH suggests that they are involved in the initial stages of the response to CIH in the NTS. The peak of astrocyte activation coincided with early CB potentiation and cardiorespiratory alterations, suggesting a crucial role for astrocytes in CIH-induced cardiorespiratory dysfunction.

## 7. Concluding Remarks and Future Perspectives

In summary, ROS signals participate in the acute process of O_2_ sensing in the CB and contribute to enhancing the chemosensory discharges during CIH. Acute hypoxia affects mitochondrial electron transport increasing ROS levels that inhibit the conductance of KvO_2_ channels, eliciting chemoreceptor cell depolarization, entry of extracellular Ca^2+^, and the release of the excitatory transmitters. In the last decade, a growing body of experimental evidence supports a novel role for CB in the pathological consequences associated with resistant hypertension, congestive cardiac failure, metabolic disorders, and obstructive sleep apnea. In these sympathetic-related diseases, an abnormally enhanced CB chemosensory responsiveness to hypoxia has been associated with sympathetic-adrenal hyperactivity. Moreover, in preclinical models of OSA, the experimental results strongly support that the hypertension and cardiorespiratory alterations induced by CIH depend on the increased production of ROS and pro-inflammatory molecules in the CB and in the NTS.

Indeed, CIH elicits cardiorespiratory alterations linked to oxidative stress, inflammation, and sympathetic overflow. The hypertension induced by long-term CIH results from the activation of neurogenic mechanisms involving the CB chemosensory potentiation and subsequent increased sympathetic activity. Enhanced CB chemosensory inputs to the NTS are essential for maintaining the cardiorespiratory and autonomic changes during short- and long-term CIH, linked to neuronal and glial activation in the NTS. The denervation of the CBs performed during short- or long-term exposure to CIH abolished the exacerbated central chemoreflex drive, reverted the activation of microglia in the NTS, reduced the increased pro-inflammatory cytokines levels in the NTS, and restored the normal arterial blood pressure, even when animals were kept under CIH. The available experimental results suggest that astrocytes participate in the initial stages of the pro-inflammatory response in the NTS. The peak of astrocyte activation coincided with early peripheral CB chemoreceptor sensitization, elevated arterial blood pressure, and enhanced ventilatory responses after 7 days of CIH, indicating a pivotal role for astrocytes in CIH-induced cardiorespiratory dysfunction in the initial stages. Therefore, targeting medullary astrocytes may offer a new treatment for high blood pressure and other cardiovascular comorbidities in OSA patients. Although reactive glial activation is an adaptive mechanism for neuronal protection, persistent CIH leads to oxidative stress and a pro-inflammatory niche in the NTS. Interestingly, acute chemogenetic inhibition of astrocytes residing in the NTS during short-term CIH restored the elevated arterial blood pressure, reduced the enhanced hypoxic ventilatory responses, and normalized the altered sympathetic function in rats exposed to 21 days of CIH. Consequently, the available evidence strongly supports the hypotheses that (1) oxidative stress induced by CIH contributes to enhancing the CB chemosensory afferent discharges to the NTS and (2) the enhanced CB afferent activity contributes to persistent hypertension and potentiated ventilatory responses to hypoxia, likely triggering neuroinflammation in the NTS.

Further studies are needed to understand how CIH-induced oxidative stress and pro-inflammatory molecules may impact the O_2_ sensing process in the CB and enhance the chemosensory discharges. Understanding how the CIH-induced CB chemoreceptor potentiation activates neurons and glial cells in brainstem cardiorespiratory centers will offer new insights into the progression of pathological complications of OSA. Targeting oxidative stress and pro-inflammatory signaling in the CB and the NTS could be a new treatment for hypertension and cardiovascular alterations in OSA patients.

## Figures and Tables

**Figure 1 antioxidants-14-00675-f001:**
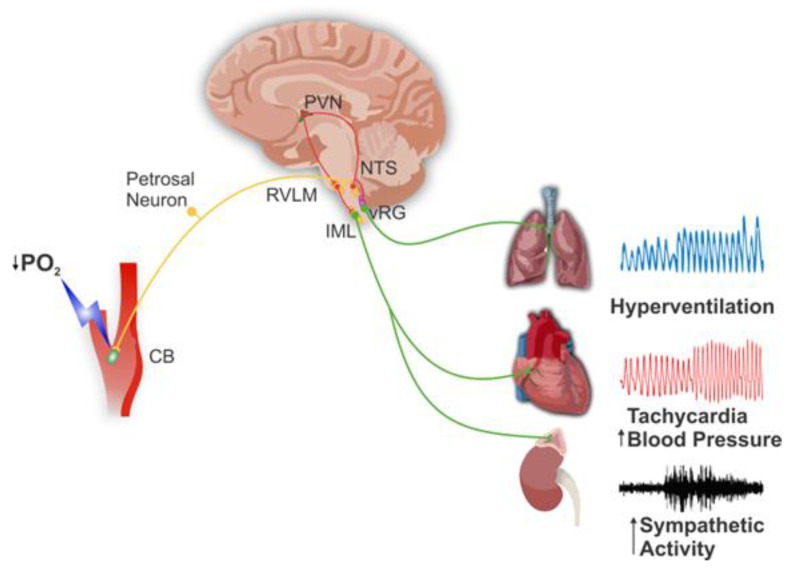
Carotid body chemoreflex pathway. During acute hypoxia, low PO_2_ stimulates the carotid body (CB) chemosensory cells releasing excitatory transmitters, which in turn increase the discharge of primary petrosal neurons that project to the nucleus of the tractus solitarius (NTS). In the NTS, second and third-order sensory neurons project to the ventral respiratory group (vRG) augmenting the ventilatory flow, to the paraventricular nucleus (PVN) that mediates hormonal and autonomic responses, and to the rostral ventrolateral medulla (RVLM) that project to the intermediolateral medulla (IML), activating the sympathetic pathway to the heart and blood vessels, which increases heart rate and arterial blood pressure. These reflexes restore arterial PO_2_ and decrease the CB chemosensory activity, thereby restoring cardiorespiratory homeostasis.

**Figure 2 antioxidants-14-00675-f002:**
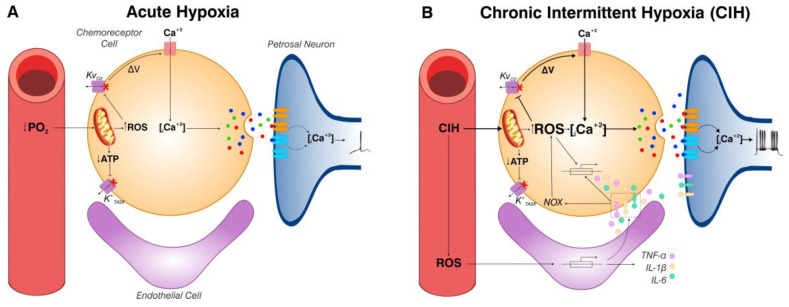
Diagram of the proposed effects and origin of ROS in acute O_2_ sensing and during CIH. (**A**): according to the most accepted hypothesis of O_2_ sensing in the CB chemoreceptor cells, acute hypoxia inhibits the mitochondrial electron transport and/or oxidative phosphorylation in chemoreceptor cells, thereby reducing ATP content or altering the ATP/ADP+Pi ratio, and increasing ROS and NADH, which in turn inhibit background K^+^ (TASK) and voltage-dependent K^+^ (KvO_2_) channels, respectively, leading to cell depolarization, Ca^2+^ entry, and release of the excitatory transmitters. (**B**): chronic intermittent hypoxia (CIH), increased ROS levels in the CB chemosensory cells (and probably in endothelial and sustentacular cells) promoting the gene expression of pro-inflammatory cytokines such as tumor necrosis factor-α (TNF-α), interleukin-1β (IL-1β), and interleukin-6 (IL-6). The oxidative stress and the pro-inflammatory niche in the CB chemoreceptor cells increase intracellular Ca^2+^, resulting in a higher release of the excitatory transmitters and enhanced chemosensory discharge.

**Figure 3 antioxidants-14-00675-f003:**
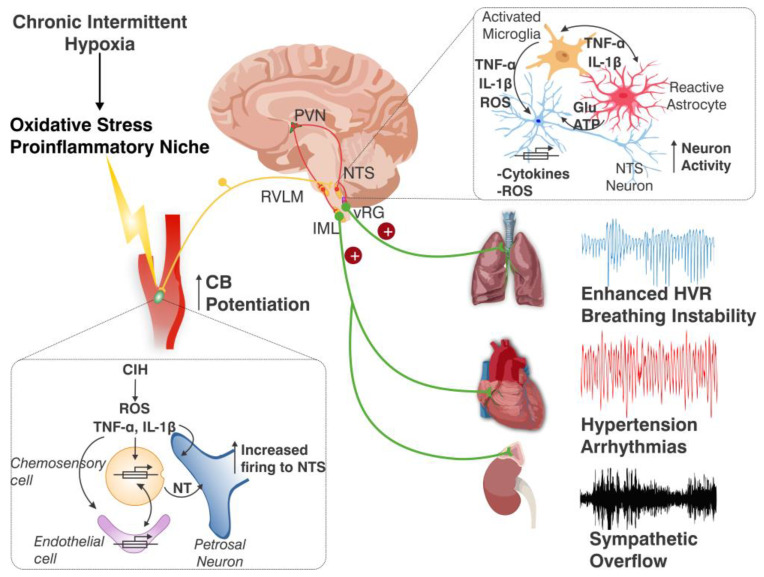
Main pathophysiological mechanisms proposed for CB chemosensory potentiation and glial and neuronal activation in the NTS during CIH. Exposure to CIH, the main component of obstructive sleep apnea triggers cardiorespiratory disturbances that contribute to perpetuating the pathological effects of CIH. In the CB, CIH produces oxidative stress and increased pro-inflammatory cytokines that contribute to enhancing the CB chemosensory discharges (lower panel). The enhanced CB chemosensory drive to the NTS promotes a phenotype shift in astrocytes and microglia generating a pro-inflammatory niche, which results in chronic activation of NTS neurons, given that cytokines and ROS augment glutamate (upper panel). The overactivation of the central CB chemoreflex pathway produces oxidative stress and neuroinflammation in brainstem cardiorespiratory and autonomic nuclei, which contributes to maintaining enhanced ventilatory responses to hypoxia, breathing instability, sympathetic overflow, arrhythmia, and hypertension. CB: carotid body, NT: excitatory neurotransmitters, vRG: ventral respiratory group, PVN: hypothalamic paraventricular nucleus, RVLM: rostral ventrolateral medulla, IML: intermediolateral medulla, IL-1β: interleukin-1β, TNF-α: tumor necrosis factor-alpha, Glu: glutamate.

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
