# Peer review of "Reactive Oxidative Species in Carotid Body Chemoreception: Their Role in Oxygen Sensing and Cardiorespiratory Alterations Induced by Chronic Intermittent Hypoxia"

_antioxidants, 2025, doi:10.3390/antiox14060675_

Round 1

Reviewer 1 Report

Title need to change.

Abstract: In the first part you should explain what is still unknown and why you want to think about this review. What is the novelty ? 

Each figure have unique title. The title can reflect your figure story. There is no proper title of your figure. Please add.

As you have only three figure, please place those figure after the introduction. It looks not good initially have figure. 

How do you create this figure ? please use proper reference. 

The figure is too general, no mechanism. Please include some proper mechanism. 

Why you did not add any sub paragraph with headline ? 

Line 273 why suddenly you used RNS ? what is the connection between ROS and RNS ?

If you use short form such as RNS, please use full form first. 

Why several full form repeated such as CB, CIH, NTS ?

Obstructive Sleep Apnea and the Carotid Body relationship is poor explanation ? need more deep mechanism and relationship. 

Line 252 -269 many time you mentioned we found ? this is review paper, why you mentioned the data ? 

Future Perspectives, there is no suitable hypothesis to support your review ? please make at least two hypothesis. 

you did not mentioned current good treatment or method properly ? 

Please modify whole manuscript again. 

same as major comment

Author Response

REVIEWER 1

We would like to thank the Reviewer for his/her valuable comments to improve the Ms.

Major comments:

Reviewer. Title need to change.

Answer. We thank Reviewers’ comment, according to his/her concern, we modified the title to: “Reactive Oxidative Species in Carotid Body Chemoreception: Their Role in Oxygen Sensing and Cardiorespiratory Alterations Induced by Chronic Intermittent Hypoxia” We hope that this title will be more descriptive.

Reviewer. Abstract: In the first part you should explain what is still unknown and why you want to think about this review. What is the novelty ?

Answer. We kindly accept Reviewers’ comments. Accordingly, we reorganized and modified the Abstract:

The carotid body (CB) senses arterial PO2, PCO2 and pH levels, eliciting reflex responses to maintain cardiorespiratory homeostasis. Chronic intermittent hypoxia (CIH), the hallmark of obstructive sleep apnea, elicits autonomic and cardiorespiratory alterations that are attributed to an enhanced CB chemosensory responsiveness to hypoxia, which in turn activates neurons and glial cells in the nucleus of the tractus solitarius (NTS). Although the CB contribution to the CIH-induced pathological alterations is well-known, the underlaying mechanisms are not fully understood. A growing body of new evidence suggests a crucial role for ROS in acute CB oxygen sensing, as well as in the potentiation of chemosensory discharge and the activation of the central chemoreflex pathway in CIH. Indeed, it has been proposed that acute hypoxia disrupts mitochondrial electron transport, increasing ROS and NADH in the chemoreceptor cells, which inhibit volt-age-gated K+ channels, producing cell depolarization, Ca2+ entry and release of excitatory transmitters. In addition, new evidence supports that the enhanced CB afferent discharge contributes to persistent CIH-induced cardiorespiratory alterations, likely triggering neu-roinflammation in the NTS. Thus, in this review we will examine the experimental evi-dence that supports the involvement of ROS in the acute O2 sensing process, and their role in the enhanced CB chemosensory discharges, the glial-related inflammation in the NTS and the cardiorespiratory alterations induced by CIH.

Reviewer. Each figure have unique title. The title can reflect your figure story. There is no proper title of your figure. Please add.

Answer. The point raised by the Reviewer is well taken. According to his/her suggestion, we added a proper title for each Figure.

Reviewer. As you have only three figure, please place those figure after the introduction. It looks not good initially have figure.

Answer. According to Reviewers’ suggestion, Figures were relocated  in the text.

Reviewer. How do you create this figure? please use proper reference.

Answer. Reviewers’ concern is well taken. All images were created by us using CorelDraw Software, without using any third-party content, hence, references to Biorender or similar sources are not applicable to our artwork. Accordingly, we added after “Author Contributions” the following sentence:

Figures: All figures were completely designed by the research group using CorelDraw Software (24.30 version), including their conceptualization, creation, and any post-production work, without using third-party content.

Reviewer. The figure is too general, no mechanism. Please include some proper mechanism.

Answer. We thank Reviewers’ comment. In this regard, we need to clarify that Figure 1 is in fact a proper mechanism, corresponding to a diagram of the CB chemosensory central pathway, which comprises several organs, including de CB, Central Nervous System, Cardiovascular and Respiratory afferents. Figures 2 and 3 were relocated in the text. We introduced proper mechanisms as the reviewers suggest in Figs 2 and 3, showing the role of oxidative stress and inflammation in the CB and peripheral organs.

Reviewer. Why you did not add any sub paragraph with headline?

Answer. This is a brief review, We believe that headlines are not necessary. We revised several Reviews published in Antioxidants, and most of them are nor subdivided with headlines.

Reviewer. Line 273 why suddenly you used RNS? what is the connection between ROS and RNS ?

Answer:  We thank Reviewers’ comment. Both ROS and RSN have been related to the CB chemosensory potentiation induced by CIH. Superoxide reacts with nitric oxide to produce peroxynitrite that nitrates tyrosyl groups in proteins altering its function through the generation of 3-NT residues. Moya et al found that the administration of Ebselen, a specific peroxynitrite scavenger, during exposure of rats to CIH for 7 days reduced the 3-NT levels in the CB, normalizing the chemosensory response to hypoxia and hypertension. (Moya et al  Intermittent Hypoxia-Induced Carotid Body Chemosensory Potentiation and Hypertension Are Critically Dependent on Peroxynitrite Formation. Oxid Med Cell Longev 2016, 2016, 9802136, doi:10.1155/2016/9802136).

Reviewer. If you use short form such as RNS, please use full form first

Answer. We carefully revised Ms. and made the appropriate changes suggested by the Reviewer.

Reviewer. Why several full form repeated such as CB, CIH, NTS ?

Answer. We carefully revised Ms. and made the appropriate changes suggested by the Reviewer.

Reviewer. Obstructive Sleep Apnea and the Carotid Body relationship is poor explanation ? need more deep mechanism and relationship.

Answer: We thank Reviewers’ comment, Accordingly, the current accepted relationship between the CB and the pathological alterations in OSA states that CIH enhances the CB chemosensory discharge to the NTS contribution to the progression of cardiorespiratory alterations, The fact that the surgical ablation of the CBs restored the altered autonomic and cardiorespiratory alterations strongly supports a role for the CB in the progression of CIH-induced alterations. Moreover, the use of antioxidants, which blocked the CB chemosensory potentiation, eliminated the alterations in preclinical models of OSA. Notwithstanding, we modified the title of this section to: Chronic Intermittent Hypoxia Enhanced Carotid Body Chemosensory discharges in Pre-clinical Models of Obstructive Sleep Apnea.

Reviewer. Line 252 -269 many time you mentioned we found ? this is review paper, why you mentioned the data ?

Answer. We thank Reviewers’ comment. We carefully revised the Ms. and made the appropriate changes to eliminate “we found.”

Reviewer: Future Perspectives, there is no suitable hypothesis to support your review ? please make at least two hypotheses.

Answer. We thank Reviewers’ comment; accordingly, we modified the Future perspective and added the hypothesis. However, in the abstract and the Introduction it is clearly stated why this review is necessary,

Reviewer: you did not mentioned current good treatment or method properly ?

Answer:  We thank Reviewers’ comment. After the section Author Contributions, we added:

Method of Revision: This work is a narrative review that provides a balanced comprehensive discussion and interpretation of the experimental published data.

Reviewer. Please modify whole manuscript again.

Answer. According to the comments of the Reviewer we modified the whole Ms.

Detail comments: same as major comment: Se previous answers.

Reviewer 2 Report

  1. This review offers a comprehensive and valuable synthesis of the role of Reactive Oxygen Species (ROS) in carotid body (CB) oxygen sensing under physiological conditions and in the context of chronic intermittent hypoxia (CIH).  
  2. The manuscript effectively links the CIH-induced potentiation of CB chemosensory activity, driven by oxidative stress and inflammation, to downstream effects in the Nucleus of the Tractus Solitarius (NTS), including neuronal and glial activation (astrocytes and microglia) and subsequent cardiorespiratory dysfunction.  
  3. While the roles of different ROS sources (mitochondria, NOX) are mentioned, the discussion could be slightly enhanced by more explicitly comparing and contrasting their specific contributions to CB alterations during acute hypoxia versus chronic intermittent hypoxia.  
  1. Figures: Please ensure final figure submissions are of high resolution for clarity in publication.  
  2. References: Please review the reference list for consistency and accuracy. Several references appear to cite future publication years (e.g., 2025) or have potential inconsistencies/typos (e.g., Ref 8 DOI, Ref 231 author initials).  
  3. English Language: While the science is clearly communicated, a final proofread by a native English speaker or professional editing service is recommended to refine sentence structure and phrasing for optimal flow and precision (e.g., minor grammatical points, phrasing like "nitro-oxidative" could potentially be clarified). A stray parenthesis was noted in line 368.

Author Response

REVIEWER 2

We would like to thank the Reviewer for his/her valuable comments to improve the Ms.

Comments for Authors

Major comments

Reviewer. This review offers a comprehensive and valuable synthesis of the role of Reactive Oxygen Species (ROS) in carotid body (CB) oxygen sensing under physiological conditions and in the context of chronic intermittent hypoxia (CIH).

The manuscript effectively links the CIH-induced potentiation of CB chemosensory activity, driven by oxidative stress and inflammation, to downstream effects in the Nucleus of the Tractus Solitarius (NTS), including neuronal and glial activation (astrocytes and microglia) and subsequent cardiorespiratory dysfunction.

Answer. We would like to thank you for the comment.

Reviewer. While the roles of different ROS sources (mitochondria, NOX) are mentioned, the discussion could be slightly enhanced by more explicitly comparing and contrasting their specific contributions to CB alterations during acute hypoxia versus chronic intermittent hypoxia.

Answer: We thank Reviewers’ comment, accordingly, we modified the title of Figure 2  To “Diagram of the proposed effects and origin of ROS in acute O2 sensing and during CIH” and add the following sentences

 Figure 2 compares the origin and effects of ROS on acute O2 sensing and during CIH. In the acute hypoxic transduction process, ROS are derived from the complex I of the electron transport chain [17,42,44]. In CIH, ROS are not only produced in complex I, but also by the NADPH oxidase [37,68].

Detail comments

Reviewer. Figures: Please ensure final figure submissions are of high resolution for clarity in publication.

Answer. In accordance with Reviewers’ suggestion, final figures were uploaded as a single high-resolution PDF (1200 dpi)

Reviewer. References: Please review the reference list for consistency and accuracy. Several references appear to cite future publication years (e.g., 2025) or have potential inconsistencies/typos (e.g., Ref 8 DOI, Ref 231 author initials).

Answer. As Reviewer requested, we check all references and made the appropriate corrections.

Reviewer. English Language: While the science is clearly communicated, a final proofread by a native English speaker or professional editing service is recommended to refine sentence structure and phrasing for optimal flow and precision (e.g., minor grammatical points, phrasing like "nitro-oxidative" could potentially be clarified). A stray parenthesis was noted in line 368.

Answer: We checked and improved the English Language in Ms. However, in case of being necessary, we will request the MDPI staff to revise the English of the final approved Ms.

Round 2

Reviewer 1 Report

I did not see change or not 

for example line 199 full form

Reviewer. Why you did not add any sub paragraph with headline?

Answer. This is a brief review, We believe that headlines are not necessary. We revised several Reviews published in Antioxidants, and most of them are nor subdivided with headlines.

I am not satisfy your reply. Did you check some journal ? I do not say about Antioxidant. Please check. 

ok

Author Response

Reviewer

I did not see change or not  for example line 199 full form

Reviewer. Why you did not add any sub paragraph with headline?

Answer: In This revised version of the Ms. we highlight all changes in red and added the sub-paragraphs with headline that the Reviewer asked,